# SPATIALLY PARALLEL CONVOLUTIONS

**Peter Jin**[*]
University of California, Berkeley
phj@eecs.berkeley.edu

**Boris Ginsburg**
NVIDIA Corporation
bginsburg@nvidia.com

**Kurt Keutzer**
University of California, Berkeley
keutzer@berkeley.edu

## ABSTRACT

The training of convolutional neural networks with large inputs on GPUs is limited by the available GPU memory capacity. In this work, we describe spatially parallel convolutions, which sidestep the memory capacity limit of a single GPU by partitioning tensors along their spatial axes across multiple GPUs. On modern multi-GPU systems, we demonstrate that spatially parallel convolutions attain excellent scaling when applied to input tensors with large spatial dimensions.

## 1 INTRODUCTION

Deep convolutional neural networks (convnets) form the basis of state-of-the-art models in computer vision. Convnets are often trained on GPUs, which have limited memory capacity; typical high-end GPUs have only 12 GB–16 GB of DRAM or HBM. Limited GPU memory presents an obstacle to training high resolution convnets on semantic segmentation and other tasks (Karras et al., 2017; Wang et al., 2017). Current frameworks implement data or model parallelism to split tensors onto multiple GPUs (Iandola et al., 2016; Krizhevsky, 2014; Dettmers, 2014). However, data and model parallelism may be insufficient. Other approaches reduce working memory size but increase computation time: these include simply executing convnets on overlapping slices of the input, as well as sophisticated checkpointing techniques (Chen et al., 2016; Gruslys et al., 2016).

To a degree, the problem of working with spatial tensors too large to fit on a single GPU is a solved problem (Micikevicius, 2009; Coates et al., 2013). In the field of stencil codes which also deals with spatial data, a tensor is spatially distributed into partitions, each of which lies on a single processing element. From the point of view of any single partition, the spatially adjacent partitions possess remote boundary data that must be communicated before performing local computation; this boundary data is called the "halo region." The implementation technique used by stencil codes is to slightly pad each partition with a "ghost zone" buffer that receives a copy of the halo region from adjacent partitions. The combined halo region copying and ghost zone padding leads to a small communication and memory overhead while preserving the total number of arithmetic operations.

In this work we demonstrate that on modern multi-GPU systems, spatially parallel convolutions on tensors with large spatial dimensions exhibit excellent multi-GPU scaling in computation time and memory usage. In a few cases, spatial parallelism yields surprising but explainable *superlinear* speedups. Our implementation is largely based on halo regions and ghost zones. Spatially parallel convolutions are additionally *adjoinable*: it is easy to backpropagate through spatially parallel convolutions for gradient-based optimization methods such as SGD.

## 2 SPATIALLY PARALLEL CONVOLUTIONS

In this section, we describe how spatially parallel convolutions work. The same approach also applies to other spatial operations, e.g. pooling. We use the following notation: $P$ = number of GPUs, $N$ = batch size, $C$ = channels, $H$ = height, $W$ = width, $K$ = conv kernel size, $D$ =

---

[*]A significant part of this work was done using DGX systems during the author's internship at NVIDIA.

dilation rate, and $R$ = halo region size. For simplicity, our exposition assumes that tensors are in $NCHW$-layout. Additionally we assume each partition is a horizontal stripe of the whole tensor.

## 2.1 FORWARD PASS

Let $w$ be the convolution kernel, let $x$ be the (unpartitioned) input tensor, and let $y$ be the (unpartitioned) output tensor. Let $x_p$ represent a partition of the tensor $x$, where the partitions are indexed by $p = 1$ to $P$ and have height $H/P$. In vectorized notation, the forward pass is a halo region *exchange* and consists of the following operations, with loop index $i = 0$ to $R - 1$:

$$x_p[:, :, H/P + i, :] \leftarrow x_{p+1}[:, :, i, :] \tag{1}$$
$$x_p[:, :, -1 - i, :] \leftarrow x_{p-1}[:, :, H/P - 1 - i, :] \tag{2}$$
$$y_p \leftarrow \text{Conv}(w, x_p). \tag{3}$$

Above, out-of-bounds array indices lie within the halo region.

## 2.2 BACKWARD PASS

Let $\Delta x \triangleq \nabla_x f$ be the gradient of the tensor $x$ with respect to a function $f$, and let $\Delta x_p$ represent a partition of $\Delta x$. The backward pass is the "transpose" of the forward pass, taking the form of a halo region *reduction* and consisting of the following, again with loop index $i = 0$ to $R - 1$:

$$\Delta w_p \leftarrow \text{ConvBwdKernel}(x_p, \Delta y_p) \tag{4}$$
$$\Delta w \leftarrow \sum_{p=1}^{P} \Delta w_p \tag{5}$$
$$\Delta x_p \leftarrow \text{TransposeConv}(w, \Delta y_p) \tag{6}$$
$$\Delta x_p[:, :, i, :] \leftarrow \Delta x_p[:, :, i, :] + \Delta x_{p-1}[:, :, H/P + i, :] \tag{7}$$
$$\Delta x_p[:, :, H/P - 1 - i, :] \leftarrow \Delta x_p[:, :, H/P - 1 - i, :] + \Delta x_{p+1}[:, :, -1 - i, :]. \tag{8}$$

The sum into $\Delta w$ is done by an all-reduce operation, which can be executed asynchronously.

## 2.3 IMPLEMENTATION

Our preliminary implementation consists of a communication phase, during which halo region data are exchanged between adjacent GPUs, and a computation phase, during which convolution routines are executed independently on each GPU; a synchronization point separates the phases. A more efficient approach would be to fuse the halo region communication with the convolution routine, taking advantage of a low latency interconnect such as NVLink; we leave this for future work.

## 3 EVALUATION

We confirmed the correctness of our spatially parallel convolution implementation by training spatially parallel ResNet architectures on ImageNet (He et al., 2016; Russakovsky et al., 2015). For example, on a spatially parallel version of ResNet-18 trained using up to 4 GPUs, we attained top-1, single-crop ILSVRC2012 validation accuracy of 69.5%.

To evaluate the effectiveness of our proposed spatially parallel convolutions, our microbenchmarks compared two approaches for a given problem size: (a) computing a convolution when the entire problem size fits on a single GPU; and (b) computing a spatially parallel convolution when the problem size is partitioned across multiple GPUs. Our evaluation platform was a Tesla V100 4xGPU system with NVLink 2.0 interconnect via the Amazon EC2 P3.8x GPU instance type. We used convolution routines from cuDNN 7 with CUDA 9.0. We exclusively ran single-precision floating point operations. We also restricted the maximum workspace size to 4 GB.

The results of our microbenchmarks are summarized in Tables 1 and 2. The numbers suggest that for easily achievable problem sizes ($H, W \gtrsim 256$) spatially parallel convolutions on multiple GPUs can achieve roughly linear speedup up to 4 GPUs compared to convolutions on a single GPU. Dilated convolutions are harder to parallelize than regular convolutions, but on larger problems ($H, W \gtrsim$

Table 1: Spatially parallel convolution (fp32): $N = 32$, $C = 64$, $K = 3$, $D = 1$, $R = 1$. For each benchmark, we show both average wall-clock time (in ms) over 1000 trials and speedup over the single-GPU case, as well as the per-GPU memory usage (in MB) and fraction of the memory used compared to the single-GPU case.

| Problem size | GPUs | Fwd. wall-clock | Bwd. wall-clock | Memory per GPU |
|---|---|---|---|---|
| $H = W = 128$ | 1 GPU | 2.56 ms ($1.0\times$) | 6.63 ms ($1.0\times$) | 256 MB ($1.00\times$) |
| | 2 GPUs | 1.52 ms ($1.7\times$) | 3.50 ms ($1.9\times$) | 134 MB ($0.52\times$) |
| | 4 GPUs | 1.23 ms ($2.1\times$) | 2.33 ms ($2.8\times$) | 69 MB ($0.27\times$) |
| $H = W = 256$ | 1 GPU | 10.02 ms ($1.0\times$) | 26.81 ms ($1.0\times$) | 1024 MB ($1.00\times$) |
| | 2 GPUs | 5.34 ms ($1.9\times$) | 11.79 ms ($2.3\times$) | 524 MB ($0.51\times$) |
| | 4 GPUs | 3.11 ms ($3.2\times$) | 6.96 ms ($3.9\times$) | 266 MB ($0.26\times$) |
| $H = W = 512$ | 1 GPU | 45.15 ms ($1.0\times$) | 126.11 ms ($1.0\times$) | 4096 MB ($1.00\times$) |
| | 2 GPUs | 20.18 ms ($2.2\times$) | 60.15 ms ($2.1\times$) | 2072 MB ($0.50\times$) |
| | 4 GPUs | 10.65 ms ($4.2\times$) | 26.76 ms ($4.7\times$) | 1044 MB ($0.25\times$) |

Table 2: Spatially parallel dilated convolution (fp32): $N = 32$, $C = 64$, $K = 3$, $D = 3$, $R = 3$.

| Problem size | GPUs | Fwd. wall-clock | Bwd. wall-clock | Memory per GPU |
|---|---|---|---|---|
| $H = W = 128$ | 1 GPU | 3.92 ms ($1.0\times$) | 11.10 ms ($1.0\times$) | 256 MB ($1.00\times$) |
| | 2 GPUs | 2.69 ms ($1.5\times$) | 6.49 ms ($1.7\times$) | 147 MB ($0.57\times$) |
| | 4 GPUs | 2.44 ms ($1.6\times$) | 5.55 ms ($2.0\times$) | 80 MB ($0.31\times$) |
| $H = W = 256$ | 1 GPU | 15.39 ms ($1.0\times$) | 43.99 ms ($1.0\times$) | 1024 MB ($1.00\times$) |
| | 2 GPUs | 8.43 ms ($1.8\times$) | 22.66 ms ($1.9\times$) | 549 MB ($0.54\times$) |
| | 4 GPUs | 5.33 ms ($2.9\times$) | 12.71 ms ($3.5\times$) | 287 MB ($0.28\times$) |
| $H = W = 512$ | 1 GPU | 61.26 ms ($1.0\times$) | 192.11 ms ($1.0\times$) | 4096 MB ($1.00\times$) |
| | 2 GPUs | 31.68 ms ($1.9\times$) | 95.42 ms ($2.0\times$) | 2120 MB ($0.52\times$) |
| | 4 GPUs | 17.51 ms ($3.5\times$) | 46.72 ms ($4.1\times$) | 1085 MB ($0.26\times$) |

512) spatially parallel dilated convolutions also scale well to 4 GPUs. In a few cases (especially the backward results), we actually observe *superlinear* speedup, which is surprising but explainable. The appearance of a superlinear speedup suggests that there is simply further room for improvement of the single-GPU convolution routines.

## 4  DISCUSSION

We described spatial parallelism for scaling convolutions on large tensors to multiple GPUs. We evaluated the performance of an implementation of spatial parallelism on a multi-GPU system and found excellent scaling in computation time and memory usage.

Spatial parallelism can be composed with other complementary approaches for parallelism and memory reduction for convolutional networks. The combination of spatial parallelism with both data and model parallelism (Gholami et al., 2018) is a design point, for which an optimum exists that minimizes the sum cost of computation and communication. Gradient checkpointing trades off between working memory size and recomputation of intermediate values in a computation graph (Chen et al., 2016; Gruslys et al., 2016); its combination with spatial parallelism further reduces per-GPU memory usage, while retaining favorable scaling on multiple GPUs.

An open source version of our preliminary implementation will be made available at:
`https://github.com/peterhj/arraydiff_cuda`

### ACKNOWLEDGMENTS

We would like to thank Ujval Kapasi for helpful discussions.

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
