# OpenReview forum: "Spatially Parallel Convolutions"
_ICLR.cc/2018/Workshop — Accept_

### Official Review · AnonReviewer1 · 2018-03-09
**memory efficient parallel convolutions**

**Rating:** 7
**Confidence:** 5

**Review:**

This paper describes spatially parallel convolutions to spatially partition tensors across multiple GPUs. The spatial parallelism provides superlinear scaling with the number of GPUs.

The speedup results are impressive. The halo "trick" reduces additional communication for getting neighboring items. The evaluation is over NVLink. Since all-reduce can be expensive, do you need to heavily rely on this technology for the scaling? An additional question is how varying the halo sizes affects performance, can you add this in Table 1? Overall, the paper presents a clever idea and provides impressive results in the evaluation.

---

### Official Review · AnonReviewer3 · 2018-03-10
**Good attempt, insufficient experiment**

**Rating:** 6
**Confidence:** 4

**Review:**

This paper engineers the cuda kernel for spatial parallelization of the convolution operator.
Since this is more of an engineering paper, there is no very innovative idea in this paper, thus it is easy to understand.
This work targets the application scenario where the neural network is very large, and can not fit a single GPU.
Pro:
This work accomplished what it intend to solve, namely parallelize the convolution kernel spatially with reasonable speed up.
This work may be useful in the future if the memory of GPU is not further increasing and yet researchers are working on larger images/videos.

Con:
The compared batch size in this paper is still 32, one could argue that in this case we can simply do data parallelization. Doing spatial parallelization will need to communicate both out of bounds data and gradient of weights, which has at most equal if not more communication compared to data parallelization. Comparison should be made in order to show the advantage of spatial than data parallelization. More experiments with large spatial size and smaller batch size are expected.

---

### Official Review · AnonReviewer2 · 2018-03-10
**Encouraging result**

**Rating:** 7
**Confidence:** 5

**Review:**

The authors address the challenge of using large inputs, e.g., high resolution images while training ConvNets on GPUs with limited memory. The proposed solution involves parallelizing convolutions across the spatial dimension, so that it can use multiple GPUs. The results show that such an operation can be efficient in practice and scales well with the number of GPUs.

Minor comments
- It would be helpful if the authors provided the baseline ResNet-18 number in the introduction
- This is just a suggestion, and please feel free to ignore - If you could also provide the final numbers for the models trained on 512x512 images, it would be nice.

---

### Decision · Program_Chairs · 2018-03-20
**ICLR 2018 Workshop Acceptance Decision**

**Decision:**

Accept

**Comment:**

Congratulations, your paper was accepted to the ICLR workshop.